# Switching Ticagrelor to 600 mg or 300 mg Clopidogrel Loading Bridge in Patients with Unstable Angina

**DOI:** 10.3390/jcm10112463

**Published:** 2021-06-02

**Authors:** Sinem Cakal, Beytullah Cakal, Zafer Güven, Aydın Rodi Tosu, Muhsin Kalyoncuoglu, Halil Ibrahim Biter, Ziya Apaydın, Ibrahim Oguz Karaca, Erdal Belen, Mehmet Mustafa Can

**Affiliations:** 1Cardiology Department, Haseki Training and Research Hospital, University of Health Sciences, 34668 Istanbul, Turkey; aydinroditosu@gmail.com (A.R.T.); mkalyoncuoglu80@gmail.com (M.K.); abrahambiter@hotmail.com (H.I.B.); ziyaapaydin@hotmail.com (Z.A.); belenerdal@gmail.com (E.B.); mehmetmustafacan@yahoo.com (M.M.C.); 2Cardiology Department, Istanbul Medipol University, 34513 Istanbul, Turkey; bcakal@hotmail.com (B.C.); drzaferguven80@gmail.com (Z.G.); oguzkaraca@hotmail.com (I.O.K.)

**Keywords:** switching protocols, coronary artery disease, de-escalation, unstable angina, antiplatelet agent, vascular disease

## Abstract

Ticagrelor is believed to be a more potent and faster antiplatelet agent compared with clopidogrel and may result in lower ischemic outcomes in patients with acute coronary syndrome. However, the best strategy of switching from ticagrelor to clopidogrel is unclear. Current guidelines advocate clopidogrel bridging with a 600 mg loading dose (LD). This study aimed to compare the safety and feasibility of switching protocols from ticagrelor to clopidogrel 600 mg or 300 mg LD in patients with unstable angina pectoris (USAP). One hundred and eighty patients with USAP undergoing adhoc percutaneous coronary intervention (PCI) received preprocedural ticagrelor 180 mg/daily. The decision to switch antiplatelet therapy to clopidogrel with either 300 mg LD or 600 mg LD at 12 h was left to the discretion of the treating physician. The primary outcome was a composite of an efficacy endpoint major adverse cardiac and cerebrovascular events (MACCEs) and a safety endpoint Bleeding Academic Research Consortium scale (BARC) (≥1). There were no differences in our composite clinical endpoint of MACCE between the two strategies, with one event occurring in each group. One patient in each group had myocardial infarction due to stent thrombosis, and the patient in the 300 mg switching group died due to stent thrombosis. No difference between the two arms was observed in terms of BARC bleeding criteria. This study showed that among USAP patients undergoing PCI, switching to clopidogrel with 300 mg LD showed no significant difference compared to 600 mg clopidogrel LD. Ticagrelor LD in ad hoc PCI and de-escalation to clopidogrel with 300 mg LD could translate to lower costs for patients with USAP without compromising safety and efficacy.

## 1. Introduction

Dual antiplatelet therapy (DAPT) with potent P2Y12 inhibitors on top of aspirin reduces ischemic events in patients following acute coronary syndrome (ACS). DAPT with novel P2Y12 inhibitors may reduce the risk of stent thrombosis at the expense of increased bleeding risk after coronary stenting [1]. Clopidogrel served as a cornerstone of DAPT for decades despite some potential limitations such as being a pro-drug with a short-lived active metabolite, relatively mild potency, slow onset of action, and a large interpatient variability of antiplatelet response [2].

Ticagrelor, known as a member of a new chemical class, cyclopentyl triazolopyrimidine, acts as a direct, oral, reversibly binding P2Y12 inhibitor with a plasma half-life of about 12 h [3]. In the PLATO trial, patients with moderate/high-risk ACS without ST segment elevation (NSTE-ACS) or ST segment elevation myocardial infarction (STEMI) planned for primary PCI received clopidogrel 75 mg daily. They had a loading dose (LD) of 300–600 mg, or ticagrelor 180 mg LD followed by 90 mg twice daily. Ticagrelor was superior to clopidogrel in ACS patients regarding death from vascular causes, myocardial infarction, or stroke without an increase in the rate of overall major bleeding [4].

Based on the 2017 ESC DAPT guidelines, ticagrelor administration (180 mg LD, 90 mg tablet twice a day) or clopidogrel instead of ticagrelor (600 mg LD, 75 mg daily dose) should be considered in patients with NSTE-ACS undergoing PCI [5]. However, a significant proportion of patients are switched back to clopidogrel mostly because of the need for oral anticoagulation, increased bleeding risk, patient preferences, the presence of side effects, and socio-economic reasons. Moreover, more than 14 percent of patients treated with ticagrelor may develop dyspnea, which may warrant discontinuation due to intolerance [6]. The appropriate approach to escalate from clopidogrel to ticagrelor is the only switch between P2Y12 inhibitors that has been investigated in a systematic review and meta-analysis powered for clinical endpoint (Figure 1) [7].

Currently, reliable evidence regarding the optimal dose approach of switching from ticagrelor to clopidogrel is unclear. Considering such an obvious gap for defining optimal loading, some prefer a higher-dose regimen (600 mg clopidogrel bolus at the time of the switch). In contrast, others consider 300 mg clopidogrel would be sufficient. We sought to evaluate safety and efficacy of giving 600 mg bolus versus 300 mg bolus clopidogrel in patients with unstable angina (USAP) who need to de-escalate from ticagrelor to clopidogrel.

## 2. Materials and Methods

The present study was a retrospective cohort study conducted in two centers in Istanbul, Turkey. One hundred and eighty patients with USAP undergoing ad hoc PCI pre-procedural ticagrelor 180 mg LD between November 2019 and December 2020. Unstable angina was defined as typical chest pain at rest or minimal exertion leading to myocardial ischemia without cardiomyocyte damage and no persistent ST-segment elevation (ECG changes, including persistent or transient ST-segment depression, T-wave inversion or pseudo normalization of T waves; or normal ECG [8]. All study subjects were over the age of 20, admitted to hospital with unstable angina pectoris (USAP), and were initially treated with ticagrelor before being switched to clopidogrel. The clopidogrel resistance testing was not performed according to current revascularization guidelines (Class IIb recommendation) [9]. The decision to switch antiplatelet therapy to clopidogrel with either 300 mg LD or 600 mg LD was at the treating physician’s discretion, and the main reasons for de-escalation were the financial burden, concerns about increased risk for bleeding, and shortness of breath. Then, 88 patients were switched to clopidogrel with 600 mg LD (600 mg group), and 92 patients were switched to 300 mg LD (300 mg group) at 12 h following ticagrelor LD. Exclusion criteria included patients under 20 years of age, thrombocytopenia (platelet count < 100.000), active bleeding, anemia (hematocrit < 30%), and receiving concomitant anticoagulant therapy. This study was approved without restrictions by the local research ethics committee. Informed consent was obtained from all subjects.

### 2.1. Study Endpoints

The primary outcome was a composite of an efficacy endpoint [major adverse cardiac and cerebrovascular events (MACCEs) defined as a composite of cardiovascular death, recurrent myocardial infarction, target vessel revascularization, and non-fatal stroke one month follow up. Median follow-up was at 8 months (IQR: 5–12) to identify late complications. The safety endpoints (clinically significant bleeding) of the 30-day Bleeding Academic Research Consortium (BARC) criteria were within one month [10]. We defined BARC II or higher as major bleeding and BARC I as minor bleeding.

### 2.2. Statistical Analysis

Continuous variables were described as mean (± standard deviation) and categorical variables as percentages. Chi-square tests for categorical variables and t-test were used for comparisons. A *p*-value less than 0.05 was considered statistically significant. Statistical analyses were performed using IBM SPSS Statistics for Windows, version 24 (IBM Corp., Armonk, NY, USA).

## 3. Results

A total of 180 patients with USAP undergoing ad hoc PCI received ticagrelor 180 mg LD. The mean age was 61.22 ± 10, and 141 of 180 (78.3%) patients were male. Fifty-seven patients (31.7%) had previous MI, 61 (33.9%) had a previous PCI. Seventy-five patients (41.7%) used concomitant DAPT, and all patients received treatment with proton pump inhibitors (PPIs). In our study, the most common PCI was performed on the LAD lesion (57.2%).

Eighty-eight patients were switched to clopidogrel with 600 mg LD (600 mg group), and 92 patients were switched to 300 mg LD (300 mg group) at 12 h according to the preference of their treating physician. Clinical, demographic characteristics and type of stents (BMS; bare-metal stent, DES; drug-eluting stent, BMS + DES) were similar in both two groups (Table 1). The reason to choose BMS over DES was mainly due to cost issues and insurance coverage.

There were no differences in the composite clinical endpoint of MACCE between the two strategies, with one event occurring in each group (Table 2). One patient in the 600 mg switch group had myocardial infarction due to stent thrombosis, and also in the 300 mg switch group, one patient suffered stent thrombosis and died. In the 600 mg group, the patient was 82 years old male with a BMS stent in LAD and a history of hypertension, diabetes, dyslipidemia, and previous MI; unfortunately suffered stent thrombosis on the 28th day after the procedure. The mechanism of this stent thrombosis was non-compliance with DAPT. The percutaneous coronary intervention was successfully performed, and full patency was achieved. In the 300 mg group, the patient was 80 years old female who had a history of only hypertension. She also had BMS implantation in the LAD, and her stent was thrombosed 72 h after the procedure. The main mechanism of her stent thrombosis was thought to be under the expansion of the stent during the procedure. PCI was attempted, but during the procedure, cardiac arrest developed, and the patient died (Figure 2). Due to rare events occurring in the BMS arm, no statistical comparison was made.

### Bleedings

BARC (Bleeding Academic Research Consortium) classification was also used as the bleeding score in terms of safety outcomes. We did not observe any difference between the two treatment strategies concerning BARC bleeding criteria (*p* = 0.17). In 600 mg group 7 patients (8%) and in 300 mg group 2, 3 patients (3.3%) had clinical bleeding. Only BARC 1 (*n* = 10) occurred in patients with bleeding (Table 3). Among the bleedings, seven patients (70%) had a femoral approach leading to groin hematoma, and three patients (30%) had radial approaches. Minor bleeding occurred more frequently in patients with femoral access (*p* = 0.001) with no difference in terms of access site between 600 mg group and 300 mg group (*p* = 0.23). Minor bleedings included groin hematoma (*n* = 6), hematuria (*n* = 1), epistaxis (*n* = 1), ecchymosis (*n* = 1), gingival bleeding (*n* = 1). Patients with BARC 1 bleeding tended to more concomitant DAPT use (*p* = 0.061). History of a previous hemorrhage did not affect BARC 1 development (*p* = 0.67). Bleeding was higher in patients who received DES over BMS might be explained by the tendency of pre-procedural DAPT use in the DES group (*p* = 0.07)

## 4. Discussion

Contrary to the traditional concept, our study indicated no advantage of clopidogrel 600 mg LD over 300 mg LD during switching from ticagrelor. Switching from a new P2Y12 receptor inhibitor to clopidogrel is common in clinical practice, but there are very limited data regarding the switch dosage. Reasons for switching to clopidogrel concern high costs associated with ticagrelor/prasugrel and bleeding caused by stronger antiplatelet agents. Generally, in-hospital switching from the novel P2Y12 receptor inhibitor to clopidogrel was reported in the registries between 5.0% and 13.6% [11].

Importantly, about half of patients in the PLATO study were switched from clopidogrel given as an LD of 300–600 mg [4]. Given the unpredictable pharmacodynamic profile of clopidogrel, 600 mg LD is recommended as the best strategy for switching from ticagrelor to clopidogrel [5]. We did not tailor the antiplatelet resistance test for monitoring the pharmacological effectiveness during switching since current revascularization guidelines do not support platelet function monitoring (Class IIb) [9,12]. De-escalation from ticagrelor to clopidogrel following initial treatment is not infrequent for ACS patients [7]. Due to ticagrelor’s faster offset, de-escalation to clopidogrel with LD is recommended in acute or chronic settings, excluding patients at high risk of bleeding [13,14].

In the SCOPE study, switching between antiplatelet therapies was observed in 2.3% in the cathlab and 3.3% at discharge. The cumulative incidence of MACE and net adverse cerebrovascular events (NACE: a combination of bleeding events and MACE) was 1.6% and 5.6%, respectively. Among patients receiving an upgrade switching, no bleeding or ischemic events occurred during the entire study period. However, downgrade switching was an independent predictor of MACE [14]. That study showed switching from clopidogrel to potent P2Y12 receptor inhibitors seems safe. In contrast, a downgrade switching in the early stages of ACS is associated with adverse clinical events in contrast to our findings.

The use of loading dose during de-escalation to clopidogrel after PCI or at hospital discharge is uncommon. Our study was conducted to determine the ideal strategy of switching from ticagrelor to clopidogrel among patients with ACS. However, it did not provide pharmacodynamic information between 600 mg LD versus 300 mg LD. In THE Capital Opti-Cross Study [15], even the bolus strategy did not increase platelet aggregation inhibition at 72 h; a lower level of platelet reactivity was present at 48 h. No difference in terms of MACE or TIMI bleeding in patients receiving LD (clopidogrel 600 mg bolus followed by 75 mg daily) or no LD (clopidogrel 75 mg daily) was observed. Similarly, we did not observe any difference between the two LD treatment strategies with respect to BARC bleeding criteria.

An increased LD of clopidogrel has resulted in faster and greater inhibition than the standard 300-mg dose. However, variability of individual platelet reactivity over time could influence efficacy and bleeding risk during PCI and surgery. A 900-mg dose of clopidogrel found no incremental benefit regarding magnitude or time to maximal platelet inhibition compared with a 600-mg dose in a randomized study. Still, the benefit was evident in both groups compared to the 300-mg dose [16]. It is unclear whether this pharmacodynamic finding could translate into increased clinical adverse events, especially with current new generation stent platforms associated with lower rates of stent thrombosis and shorter mandatory DAPT requirement. In line with this, we did not find inconsistency in the composite clinical endpoint of MACCE or BARC bleeding scores between the two strategies, with one event occurring in each group.

Studies with crossover design with various clinical settings assessed the pharmacodynamic effect of switching from a new P2Y12-receptor inhibitor to clopidogrel. These tailored antiplatelet switching strategies were associated with a reduction in platelet inhibition effects and an increase in high on-treatment platelet reactivity [16]. Although some studies have reported a lower potential risk of bleeding complications once switched to clopidogrel, caution is needed to properly interpret these results due to the small sample size of the studies above. The results of the SWAP-4 study, including patients with stable angina, showed that switching from ticagrelor to clopidogrel with a maintenance dose of 75 mg led to greater increase in platelet reactivity compared to 600 mg LD on the basis of pharmacodynamic measure with no difference giving clopidogrel at 12 or 24 h [17].

Although the optimal timing of switching after the last dose of ticagrelor is unknown, the SWAP-3 study showed changing from prasugrel to ticagrelor at 12 h could initially increase platelet inhibition at 2 h without any drug interaction, which suggests the presence of a residual effect of ticagrelor on the P2Y12 receptor [18].

While previous studies [12,15] used a 600 mg clopidogrel loading dose with success, a 300 mg clopidogrel bolus dose, as shown in the current study, seems reasonable given the underlying inhibition of platelet aggregation that persists 12 h after the last dose of ticagrelor.

The main limitation of this study was a retrospective cohort study conducted in two centers, and the small sample size, which may limit the power to detect differences in clinical outcomes. Controls are missing in some experiments. No evaluation of the pharmacodynamic effects of the switching was another limitation; however, obtaining clinical outcomes would be a more pragmatic answer to whether switching is applicable. Relatively more frequent use of BMS over DES due to insurance coverage might have also affected the outcomes. A shorter duration of follow-up might limit the longer effect of switching. Additionally, another limitation was that only USAP patients were included in the study. Larger better-synchronized studies are needed.

## 5. Conclusions

This study suggests that among USAP patients undergoing ad hoc PCI, de-escalation to 300 mg clopidogrel showed no significant difference in MACCE and BARC bleeding scores compared to de-escalation with 600 mg LD of clopidogrel. Ticagrelor LD and de-escalation to clopidogrel with 300 mg LD could translate to lower costs for patients with USAP without compromising safety and efficacy. Larger prospective randomized controlled trials could reveal further evidence on the impact of switching.

## Figures and Tables

**Figure 1 jcm-10-02463-f001:**
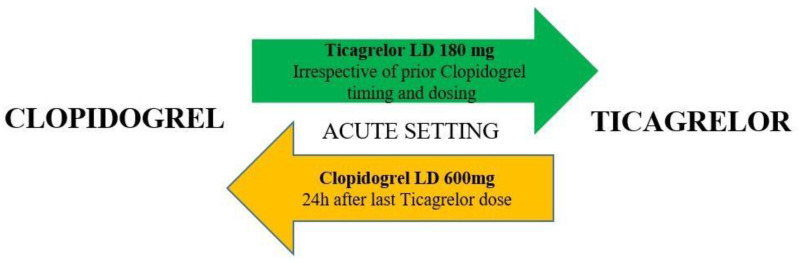
Algorithm for switching between oral P2Y12 inhibitors in the acute setting. LD = loading dose. Color-coding refers to the ESC Classes of Recommendations: green = Class I; orange = Class IIb (ESC Dual Antiplatelet Therapy Guidelines 2017).

**Figure 2 jcm-10-02463-f002:**
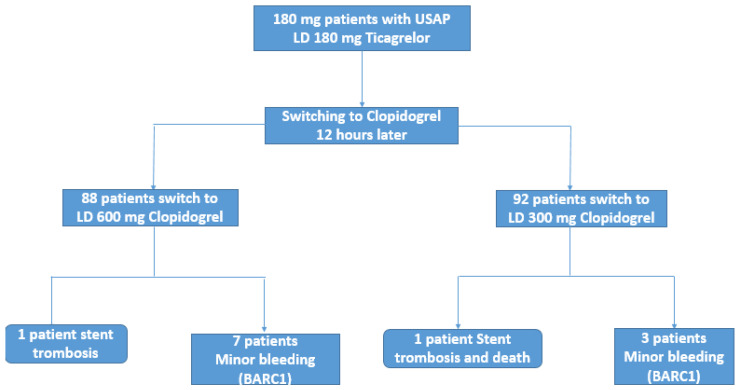
Diagram depicting the patient pathway and outcomes for switching from Ticagrelor to Clopidogrel with either 300 mg or 600 mg loading dose during a one-month follow-up.

**Table 1 jcm-10-02463-t001:** Baseline characteristics of the study population.

	Total Number (*n* = 180)	Clopidogrel Switched with 600 mg Bolus (*n* = 88)	Clopidogrel Switched with 300 mg Bolus (*n* = 92)	*p*
Age (year)	61.22 ± 10	60.14 ± 9.2	62.25 ± 10	0.063
Sex (male)	141 (78.3%)	67 (76.1%)	74 (80.4%)	0.484
Hemoglobin (mg/dL)	13.3 ± 1.6	13.4 ± 1.45	13.2 ± 1.7	0.061
Creatinine (mg/dL)	1 ± 0.53	1.03 ± 0.5	0.97 ± 0.5	0.86
Platelet (mg/dL)	242.2 ± 84	236.94 ± 55	245.29 ± 105	0.116
Fasting glucose (mg/dL)	138.26 ± 60.4	129.91 ± 62	146.25 ± 58	0.314
LDL (mg/dL)	132.62 ± 34.6	120.43 ± 32	144.2 ± 32	0.585
HDL (mg/dL)	45.6 ± 9	45.72 ± 10.6	45.5 ± 7	0.001
Triglyceride (mg/dL)	142.7 ± 34	141.75 ± 36.1	143.75 ± 32	0.63
HPR ion	130 (72.2%)	62 (70.5%)	68 (73.9%)	0.5
Diabetes mellitus	83 (46.1%)	39 (44.3%)	44 (47.8%)	0.6
Dyslipidemia	125 (69.4%)	59 (67%)	66 (71.7%)	0.49
Smoker	88 (48.9%)	40 (45.5%)	48 (52.2%)	0.36
Previous MI	57 (31.7%)	27 (30.7%)	30 (32.6%)	0.78
Previous CABG	6 (3.3%)	3 (3.4%)	3 (3.3%)	0.95
Previous PCI	61 (33.9%)	29 (33%)	32 (34.8%)	0.79
Previous Hemorrhage	3 (1.7%)	0 (0%)	3 (3.3%)	0.088
Concomitant DAPT use	75 (41.7%)	39 (44.3%)	36 (39.1%)	0.48
Number of stents				0.202
1	160 (88.9%)	77 (87.5%)	83 (90.2%)	
2	17 (9.4%)	8 (9.1%)	9 (9.8%)	
3	3 (1.7%)	3 (3.4%)	0 (0%)	
Punction				0.23
Radial	134 (74.4%)	62 (70.5%)	72 (78.3%)	
Femoral	46 (25.6%)	26 (28.7%)	20 (19.8%)	
LAD	103 (57.2%)	54 (61.3%)	49 (53.3%)	0.34
CX	41 (22.8%)	19 (21.6%)	22 (23.9%)	0.710
RCA	51 (28.3%)	24 (27.3%)	27 (29.3%)	0.75
Type of stent				0.181
BMS	77 (42.8%)	34 (38.6%)	43 (46.7%)	
DES	98 (54.4%)	53 (60.2%)	45 (48.9%)	
DES + BMS	5 (2.8%)	1 (1.1%)	4 (4.3%)	

LDL, low-density lipoprotein; HDL, high-density lipoprotein; MI, myocardial infarction; CABG, coronary artery bypass graft; PCI, Percutaneous Coronary Intervention; DAPT, dual antiplatelet therapy; LAD, left anterior descending artery; Cx, circumflex artery; RCA, right coronary artery; BMS, bare metal stent; DES, drugeluting stent.

**Table 2 jcm-10-02463-t002:** Study outcomes.

	Number of First Encountered Event		
	Underwent Switching to Clopidogrel 600 mg Bolus (*n* = 88)	Underwent Switching to Clopidogrel 300 mg Bolus (*n* = 92)	*p*
**MACCE**			
Target vessel revascularization	1	1	NA
Myocardial infarction	1	1	NA
All-cause mortality	0	1	NA
Stroke	0	0	NA
Stent Thrombosis	1	1	NA
**Clinical significant bleeds**			
BARC 1	7 (8%)	3 (3.3%)	0.169

BARC, Bleeding Academic Research Consortium; MACCE, major adverse cardiac and cerebrovascular events.

**Table 3 jcm-10-02463-t003:** Detailed description of bleeding (BARC1).

	BARC 1 (*n* = 10)	No Bleeding (*n* = 170)	*p*
Femoral approach, *n* (%)	7 (70%)	39 (22.9%)	0.001
Concomitant DAPT use, *n* (%)	7 (70%)	68 (40%)	0.061
History of previous hemorrhage, *n* (%)	0	3 (1.8%)	0.672
Type of stent, *n* (%)			0.052
BMS	1 (10%)	76 (44.7%)	
DES	8 (80%)	90 (52.9%)	
DES + BMS	1 (10%)	4 (2.4%)	

BARC, Bleeding Academic Research Consortium; DAPT, dual antiplatelet therapy; BMS, bare metal stent; DES, drug-eluting stent.

## Data Availability

The data presented in this study are available on request from the corresponding author. The data are not publicly available to protect the privacy of the patients.

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
