# Peer review of "Switching Ticagrelor to 600 mg or 300 mg Clopidogrel Loading Bridge in Patients with Unstable Angina"

_jcm, 2021, doi:10.3390/jcm10112463_

Round 1
Reviewer 1 Report
Thank you very much for this topic , I only have one concern ...why did you shift from ticagrelor to clopidogrel
Author Response
Dear Reviewer 1
Thank you very much for your kind evaluation of this study.
Line 80-83: The decision to switch antiplatelet therapy to clopidogrel with either 300 mg LD or 600 mg LD was at the discretion of the treating physician and the main reasons for de-escalation were financial burden, concerns about increased risk for bleeding and shortness of breath
Yours sincerely

Reviewer 2 Report
- How many patients had an indication for oral anticoagulation with a DOAC or warfarin. Did that seem to affect primary outcome or bleeding as stated in Table 3
- Speaking of the two patients who had in-stent thrombosis, are we able to identify the mechanism of this stent thrombosis? (i.e: late initial loading of DAPT, clopidogrel resistance).
- Was there a reason to choose BMS over DES or was it mainly for cost issues.
- Considering both complications occured in the BMS arm, are we able to provide any advanced statistics ( chi square..) to see if there was an association with BMS use alone.
- Is there a reason why bleeding was higher in patients who received DES over BMS ( Table 3)
- Was the study design supported by clopidogrel resistance testing in both arms to see by which method are therapeutic levels reached fastest.
- For how long were the patients followed after discharge to identify late complications?
Author Response
Dear Reviewer 2
Thank you very much for your kind evaluation of this study.
Line 85-88 Exclusion criteria included patients under 20 years of age, thrombocytopenia (platelet count <100.000), active bleeding anemia (hematocrit <30 %) and patients receiving oral anticoagulant therapy with a DOAC or warfarin
Line 125-133 : In 600mg group, patient was 82 years old male with a BMS stent in LAD and a history of hypertension (HT), diabetes, dyslipidemia, and previous MI unfortunately suffered stent thrombosis at 28th day after the procedure. The mechanism of this stent thrombosis was non-compliance with DAPT. Percutaneous coronary intervention was successfully performed and full patency was achieved. In 300mg group, patient was 80 years old female who had a history of only HT. She had also BMS implantation in the LAD and her stent was thrombosed 72 hours after the procedure. The main mechanism of her stent thrombosis was thought to be underexpansion of the stent during the procedure. PCI was attempted, but during the procedure cardiac arrest developed and the patient died
Line 114-115 : The reason to choose BMS over DES was it mainly for cost issues and insurance coverage.
Line 134: Due to rare events occuring in the BMS arm, no statistically comparison was made
Line 152-154: Bleeding was higher in patients who received DES over BMS might be explained by the tendency of pre-procedural DAPT use in DES group (p=0.07)
Line 79-80: The clopidogrel resistance testing was not performed according to current revascularization guidelines (Class IIb recommendation) [10].
Line 91-95: The primary outcome was a composite of an efficacy endpoint [major adverse cardiac and cerebrovascular events (MACCEs)] defined as a composite of cardiovascular death, recurrent myocardial infarction, target vessel revascularization and non-fatal stroke at 72 hours and one month follow up. Median follow up was 8 months (IQR: 5-12) to identify late complications
Yours sincerely

Reviewer 3 Report
This paper by Cakal et al. aimed to investigate the efficacy and safety of a 300mg vs 600mg switching protocol from ticagrelor to clopidogrel. They conclude that de-escalation using a 300mg is safe in patients with unstable AP.
I have a few comments:
1) line 33: this could be simply changed to "potent P2Y12 inhibitors" because both prasugrel and ticagrelor (although one is a thienopyridine-type and the other is not) can be used in ACS patients.
2) line 41: insert reference
3) line 47: "... ticagrelor was superior to clopidogrel..." in regard to what parameter/outcome?
4) line 56-60: A trial is mentioned, yet the reference includes a meta-analysis?
5) line 73: The study was conducted in two centers. Please add city and country.
6) Please add a short definition of unstable AP.
7) At what time point was the ticagrelor switched to clopidogrel? Please give details.
8) Why were patients < 20 years of age exluded? Why not < 18 years of age?
9) Did any of the patients receive concomitant anticoagulation? How many patients received treatment with PPIs?
10) In your definition of MACCE, the outcome "non-fatal stroke" is missing. For a sensitivity analysis, you could also add bleeding according to the TIMI bleeding score.
11) The primary outcome measured at 30 days is fine. However, with special regard to the switching protocol, we are particularly curious about any ischemic events during the first days. You could add this information/outcome.
12) It seems more practical to name the groups "600mg group" instead of "group 1".
13) It seems to me as quite a lot of patients received BMS instead of DES. This should be discussed or added to the limitations section.
14) line 156-158: please add a reference.
15) line 172: please detail the outcome of the mentioned trial.
16) another limitation: to a certain degree only applicable to unstable AP (no STEMI, no NSTEMI)!
17) line 206: the abbreviation HPR should be explained.
18) Overall, the discussion section at times seems incoherent and unfocused.
19) Overall extensive improvement of the english language is needed
Author Response
Dear Reviewer 3
Thank you very much for your kind evaluation of this study.
Line 33-34: Dual antiplatelet therapy (DAPT) with potent P2Y12 inhibitors on top of aspirin reduces ischemic events in patients following acute coronary syndrome.
Line 42: Ticagrelor is a member of a new chemical class, cyclopentyltriazolopyrimidine. It is a direct, oral, reversible, non-competitive P2Y12 inhibitor with a plasma half-life turnaround of about 12 h [Reference 4=Valgimigli, M.; Bueno, H.; Byrne, R.A.; Collet, J.P.; Costa, F.; Jeppsson, A.; Jüni, P.; Kastrati, A.; Kolh, P.; Mauri, L.; et al. 2017 ESC focused update on dual antiplatelet therapy in coronary artery disease developed in collaboration with EACTS: The Task Force for dual antiplatelet therapy in coronary artery disease of the European Society of Cardiology (ESC) and of the European Association for Cardio-Thoracic Surgery (EACTS). European heart journal 2018, 39, 213-260, doi:10.1093/eurheartj/ehx419].
Line 45-47: Ticagrelor was superior to clopidogrel in ACS patients, in regard to death from vascular causes, myocardial infarction, or stroke without an increase in the rate of overall major bleeding [3]
Line 55-57: The transition strategy from clopidogrel to ticagrelor is the only switch between P2Y12 inhibitors that has been investigated in a systematic review and meta-analysis powered for clinical endpoint
Line 74-75: The present study was a retrospective cohort study, conducted in two centers at Istanbul, Turkey
Line 69-72: Patients with USAP was classified according to exhibition of acute typically chest pain with myocardial ischaemia without cell damage and no persistent ST-segment elevation (ECG changes including persistent or transient ST-segment depression, T-wave inversion or pseudonormalization of T waves; or normal ECG)
Line 83-85: 88 patients were switched to clopidogrel with 600 mg LD (600 mg group) and 92 patients were switched to 300 mg LD (300 mg group) at 12 hours following ticagrelor LD
Line 85-88: Exclusion criteria included patients under 20 years of age, thrombocytopenia (platelet count <100.000), active bleeding anemia (hematocrit <30 %) and being under oral anticoagulant therapy with a direct oral anticoagulants or warfarin
Line 108: All patients received treatment with proton pump inhibitors (PPIs)
Line 91-97: The primary outcome was a composite of an efficacy endpoint [major adverse cardiac and cerebrovascular events (MACCEs)] defined as a composite of cardiovascular death, recurrent myocardial infarction, target vessel revascularization and non-fatal stroke at 72 hours and one month follow up. Median follow up was 8 months (IQR: 5-12) to identify late complications. The safety endpoints (clinically significant bleeding) of 30-day Bleeding Academic Research Consortium (BARC) criteria within one month [8]. We defined BARC II or higher as major bleeding and BARC I as minor bleeding. We did not use TIMI bleeding score. We used BARC bleeding criteria since BARC is more frequently use in current trials
Line 129-132: In 300mg group, patient was 80 years old female who had a history of only HT. She had also BMS implantation in the LAD and her stent was thrombosed 72 hours after the procedure. The main mechanism of her stent thrombosis was thought to be underexpansion of the stent during the procedure
We changed the groups "600mg group" instead of "group 1 and "300mg group" instead of "group 2
Line 114-115: The reason to choose BMS over DES was it mainly for cost issues and insurance coverage
Line 227-228: Relatively more frequent use of BMS over DES due to insurance coverage might have affected the outcomes
Line 167-169: Given the unpredictable pharmacodynamic profile of clopidogrel, 600 mg LD is recommended as the best strategy for switching from ticagrelor to clopidogrel [4=Valgimigli, M.; Bueno, H.; Byrne, R.A.; Collet, J.P.; Costa, F.; Jeppsson, A.; Jüni, P.; Kastrati, A.; Kolh, P.; Mauri, L.; et al. 2017 ESC focused update on dual antiplatelet therapy in coronary artery disease developed in collaboration with EACTS: The Task Force for dual antiplatelet therapy in coronary artery disease of the European Society of Cardiology (ESC) and of the European Association for Cardio-Thoracic Surgery (EACTS). European heart journal 2018, 39, 213-260, doi:10.1093/eurheartj/ehx419]
Line 175-180: Switching of antiplatelet therapies observed in 2.3% in the cathlab and 3.3% at discharge, in SCOPE study. The cumulative incidence of MACE and net adverse cerebrovascular events (NACE: a combination of bleeding events and MACE), was 1.6% and 5.6%, respectively. Among patients receiving an upgrade switching, no bleeding or ischemic events occurred during the entire study period. However, downgrade switching was an independent predictor of MACE [13]
Line 229-230: Also, another limitation was that only USAP patients were included in the study.
Line 207: High on-treatment platelet reactivity (HPR)
Discussion was revised
English editing was done
Yours Sincerely

Round 2
Reviewer 3 Report
1) Please name the exact primary endpoint in the abstract (e.g. BARC 2 and higher). Also add when the primary endpoint was measured.
2) line 57-58: You mention the PLATO trial, but I'm not sure in what context. Please explain. Also, the sentence is not easily understandable.
3) line 68-72: You have added a definition of USAP but the sentence is very confusing. Please revise. Also, this paragraph should be moved to the methods section. Please add a reference for the definition.
4) Patients received the clopidogrel LD at 12h after the ticagrelor LD. Why not at 24h? The guidelines that are mentioned recommend switching at 24h. Also, at what timepoint was the ticagrelor LD given? Please provide details.
5) study endpoints, line 90-97: You can't choose two primary endpoints (measured at 72h and one month). Pick one and set the other as a secondary endpoint.
6) It is written that the primary endpoint was assessed at 72h and at one month but this differentiation is not followed through in the results section. Please explain/revise.
7) line 152-154: How was the pre-procedural DAPT use in the DES group different than in the BMS group? Please explain.
8) Platelet function testing was not performed because it is not recommended by current guidelines. This is technically true but platelet function testing was performed in the SWAP-4 study and is still of great interest in such a setting. I do not think that this a valid argument against platelet function testing. Quite the opposite is the case - additional data about platelet function at certain pre-defined timepoints before and after the DAPT switch would be interesting.
9) Some abbreviations seem redundant (e.g. HT for hypertension, PI for platelet inhibition and HPR for high on-treatment platelet reactivity).
10) Figure 2: Please mention, at what timepoint the outcome (primary outcome?) was assessed. 72h or one month?
11) Did any patients in each group receive concomitant anticoagulation (e.g. phenprocoumon or low molecular weight heparin)?
12) The manuscript still lacks correct english grammar. Please revise.
Author Response
Dear Reviewer 3
Thank you very much for your kind evaluation of this study.
- Please name the exact primary endpoint in the abstract (e.g. BARC 2 and higher). Also add when the primary endpoint was measured.
Line 20: BARC Score ≥ 1 was defined as an endpoint in Abstract
- Line 57-58: You mention the PLATO trial, but I'm not sure in what context. Please explain. Also, the sentence is not easily understandable.
We removed the sentece in Lines 57-58 since it does not make sense actually as you emphasized and similar phrase was already used in Lines 171-172
- line 68-72: You have added a definition of USAP but the sentence is very confusing. Please revise. Also, this paragraph should be moved to the methods section. Please add a reference for the definition.
Line 77-80: We added reference, also changed it to ‘Methods’ section and reshaped the sentence to make it more understandable.
- Patients received the clopidogrel LD at 12h after the ticagrelor LD. Why not at 24h? The guidelines that are mentioned recommend switching at 24h. Also, at what timepoint was the ticagrelor LD given? Please provide details.
In SWAP-4 trial ticagrelor to clopidogrel switching was evaluated according to the pharmacodynamic profiles and either 12 or 24 hours switching was found to be effective and both switching methods were advised as a reasonable protocol. We mentioned this in Lines 217-221. Ticagrelor loading was given pre-procedural ‘Line 76’
- study endpoints, line 90-97: You can't choose two primary endpoints (measured at 72h and one month). Pick one and set the other as a secondary endpoint.
‘At 72 hours’ end point’ was removed in Line 97
- It is written that the primary endpoint was assessed at 72h and at one month but this differentiation is not followed through in the results section. Please explain/revise.
We removed the 72 hours endpoint and provided one month follow up since number of endpoint is rare. Line 97-98
- line 152-154: How was the pre-procedural DAPT use in the DES group different than in the BMS group? Please explain.
These group consisted of patients with a recent history of PCI procedures
- Platelet function testing was not performed because it is not recommended by current guidelines. This is technically true but platelet function testing was performed in the SWAP-4 study and is still of great interest in such a setting. I do not think that this a valid argument against platelet function testing. Quite the opposite is the case - additional data about platelet function at certain pre-defined timepoints before and after the DAPT switch would be interesting.
This could have helped the results of our study to be more powerful the study design did not included such a protocol due to retrospective nature of the study
- Some abbreviations seem redundant (e.g. HT for hypertension, PI for platelet inhibition and HPR for high on-treatment platelet reactivity).
We changed the abbreviations Lines 130-135-200-204-214-215
- Figure 2: Please mention, at what timepoint the outcome (primary outcome?) was assessed. 72h or one month?
The primary and safety outcomes were one-month Line 144
- Did any patients in each group receive concomitant anticoagulation (e.g. phenprocoumon or low molecular weight heparin)?
We excluded patients receiving concomitant anticoagulant therapy (e.g. phenprocoumon or low molecular weight heparin) Line 91-92
- The manuscript still lacks correct english grammar. Please revise.
We revised English grammar

This manuscript is a resubmission of an earlier submission. The following is a list of the peer review reports and author responses from that submission.